# Language-Bias-Resilient Visual Question Answering via Adaptive Multi-Margin Collaborative Debiasing

**Huanjia Zhu**
Beijing Institute of Technology, Zhuhai
bvyih3@gmail.com

**Shuyuan Zheng**
The University of Osaka
zheng@ist.osaka-u.ac.jp

**Yishu Liu**
Harbin Institute of Technology, Shenzhen
liuyishu@stu.hit.edu.cn

**Sudong Cai**[*]
Beijing Institute of Technology, Zhuhai
caisudong.ai@gmail.com

**Bingzhi Chen**[*]
Beijing Institute of Technology, Zhuhai
chenbingzhi@bit.edu.cn

## Abstract

Language bias in Visual Question Answering (VQA) arises when models exploit spurious statistical correlations between question templates and answers, particularly in out-of-distribution scenarios, thereby neglecting essential visual cues and compromising genuine multimodal reasoning. Despite numerous efforts to enhance the robustness of VQA models, a principled understanding of how such bias originates and influences model behavior remains underdeveloped. In this paper, we address this gap through a comprehensive empirical and theoretical analysis, revealing that modality-specific gradient imbalances, which originate from the inherent heterogeneity of multimodal data, lead to skewed feature fusion and biased classifier weights. To alleviate these issues, we propose a novel Multi-Margin Collaborative Debiasing (MMCD) framework[2], which adaptively integrates frequency-aware, confidence-aware, and difficulty-aware angular margins with a dynamic, difficulty-aware contrastive learning mechanism to reshape decision boundaries under biased training conditions. Extensive experiments across multiple challenging VQA benchmarks confirm the consistent superiority of our proposed MMCD over state-of-the-art baselines in combating language bias.

## 1 Introduction

Visual Question Answering (VQA) has emerged as a challenging task that blends computer vision and natural language processing to provide answers to natural language questions about images. The core difficulty of VQA models lies in their ability to reason multimodally, combining visual information from images with language patterns in questions. Recent progress [36, 29, 42, 7, 4, 38] in deep learning has enhanced the capabilities of VQA models. However, studies have shown that networks still suffer from language bias [15, 6, 9, 17, 4, 29, 36], where the model learns spurious correlations between questions and answers. This bias occurs when models overly rely on common patterns in questions and answers, neglecting crucial visual information. These models often perform well

---

[*]Corresponding authors: Bingzhi Chen and Sudong Cai.
[2]Code is available at https://github.com/bvyih3/2025-NIPS-MMCD

39th Conference on Neural Information Processing Systems (NeurIPS 2025).

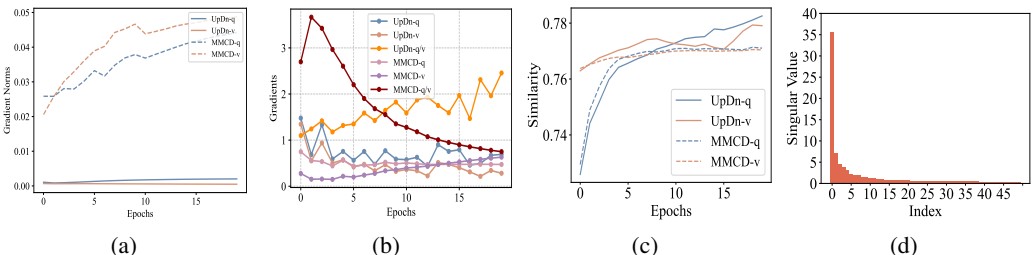

(a)                (b)                (c)                (d)

Figure 1: (a) The gradient norms across modalities differ considerably, reflecting an imbalance in how learning signals are propagated. (b) This imbalance leads to modality-specific optimization deviations, where the question modality of baseline disproportionately accumulates gradient updates, thereby amplifying its influence. (c) As a result, the fused representation becomes skewed, with question features occupying a dominant share of the multimodal space and suppressing contributions from visual features. (d) Furthermore, the classifier weights reveal directional bias: the singular value spectrum is highly uneven, suggesting that the model primarily aligns with directions that capture biased cues while overlooking secondary directions that encode meaningful information.

in standard in-distribution (ID) evaluation settings [15] but fail to generalize to out-of-distribution (OOD) data [1, 13, 21], where the distribution of answers differs from the training set.

To address this challenge, existing approaches can be grouped into three principal paradigms: ensemble-based [32, 6, 12, 28, 17, 18, 29, 36], augmentation-based [9, 25, 41, 45], and feature space-based [16, 4, 47] methods. Ensemble-based approaches [32, 6, 12] augment the base model with auxiliary bias estimators, such as question-only branches, to identify and subtract spurious correlations. Augmentation-based techniques synthesize counterfactual data [9, 25, 45] or inject negative samples [35, 41] via heuristic rules to rebalance answer distributions. Feature space-based methods impose angular margins [16, 4, 47] on the hyperspherical embedding to enforce class separability and suppress dominant priors. Despite notable progress in bias reduction [17, 4, 29, 36], a fundamental question remains unexplored: **How are language biases formed?**

Understanding the genesis of language bias is essential for advancing debiasing techniques. In this work, we conduct a comprehensive investigation into bias formation. First, we identify a modality gradient optimization deviation (see Fig. 1(b), Section 2.1), where the image modality is under-optimized and the question modality is over-optimized. Second, we observe a feature fusion component deviation (see Fig. 1(c), Section 2.2), in which question features dominate the joint representation and image features are marginalized. These phenomena culminate in directional deviation of the classifier weights (see Fig. 1(d), Section 2.2), amplifying primary (question-driven) directions and attenuating secondary (vision-driven) axes.

To mitigate language bias, we examine the efficacy of margin-based objectives, which refine model training by manipulating the angle between normalized feature and classifier weight vectors on the unit hypersphere. Despite their empirical success across diverse domains [5, 14, 27, 37, 8, 22, 24, 16, 4, 47], a fundamental question remains unanswered: **Why does margin mechanisms work?** Grounded in our analysis of bias, we show that margin penalties effectively balance modality gradient optimization (see Section 3.1), homogenize fused feature components (see Section 3.2), and equalize classifier weight directions (see Section 3.3), counteracting the skew induced by data heterogeneity.

Motivated by these findings, we propose an adaptive **M**ulti-**M**argin **C**ollaborative **D**ebiasing paradigm called "**MMCD**", which aims to reshape discriminative class boundaries. Specifically, the proposed MMCD incorporates two well-established mechanisms, i.e., **M**ulti-grained **A**daptive **M**argins (**MAM**) and **D**ifficulty-aware **C**ontrastive **L**earning (**DCL**). Specifically, MAM augments class discriminability through three adaptive angular penalties: frequency-aware, confidence-aware, and difficulty-aware margins. **1)** Frequency-aware margins counteract class imbalance by assigning larger angular offsets to underrepresented answers. **2)** Confidence-aware margins leverage logits as a proxy for sample uncertainty, dynamically tightening boundaries around ambiguous instances. **3)** Difficulty-aware margins employ an instance difficulty estimator constructed from per-sample learning speed and classification margins to further calibrate penalties at the sample level. Furthermore, DCL empowers the network to facilitate intra-class compactness and inter-class separation by incorporating the introduced difficulty model into the supervised contrastive framework. To the best of our knowledge, this is *the first attempt* to investigate the formation mechanisms of language bias in VQA.

## 2   How are Language Biases Formed?

Robust VQA has attracted extensive research interest in recent years, yielding a rich suite of debiasing techniques. Yet, the origin of language bias itself remains uncharted. In this section, we undertake a systematic investigation, combining empirical measurements with theoretical insights to trace the full trajectory by which spurious question–answer priors become entrenched in VQA models.

To begin with, we first establish some task-specific preliminaries. Given an image $v \in \mathcal{V}$ and a question $q \in \mathcal{Q}$, the goal of VQA is to predict an answer $a \in \mathcal{A}$ by optimizing a mapping $f : \mathcal{V} \times \mathcal{Q} \to \mathbb{R}^C$, where $f(v, q)$ is the logits for candidate answers, $C = |\mathcal{A}|$ is the number of candidate answers. Standard VQA frameworks [17, 18, 4, 47] typically adopt a four-stage pipeline: (1) image encoder $e_v$, (2) question encoder $e_q$, (3) multimodal fusion module $g$ which fuses unimodal features to generate joint representations $\mathcal{R}$, (4) classifier $c$ which maps $\mathcal{R}$ to answer logits with learnable weights $\mathcal{W}$. The VQA problem is formulated as:

$$f(v, q) = c(g(e_v(v), e_q(q))). \tag{1}$$

**Cross-Entropy Loss.**   The VQA model is trained by minimizing the Cross-Entropy (*CE*) loss:

$$\mathcal{L}_{\text{CE}} = \sum_{i=1}^{|\mathcal{A}|} -a_i \log \frac{\exp(f_i)}{\sum_{j=1}^{|\mathcal{A}|} \exp(f_j)}. \tag{2}$$

where $f_i$ is the logit for the $i$-th answer. We adopt UpDn [2] as our baseline. The feature fusion strategy $g$ is the Hadamard product $\odot$, thus $\mathcal{R} = \mathcal{R}_q \odot \mathcal{R}_v$, where $\mathcal{R}_q$ and $\mathcal{R}_v$ are question features and image features, respectively.

### 2.1   Modality Gradient Optimization Deviation

Generally, models store the critical information for unimodal features captured from training data in their encoder weights. We denote the weights of $e_q$ and $e_v$ as $\mathcal{W}_q$ and $\mathcal{W}_v$, respectively. We calculate the gradient of encoder weights with respect to the loss $\mathcal{L}$:

$$\nabla_{\mathcal{W}_q} \mathcal{L} = \frac{\partial \mathcal{L}}{\partial \mathcal{R}} \cdot \frac{\partial \mathcal{R}}{\partial \mathcal{R}_q} \cdot \frac{\partial \mathcal{R}_q}{\partial \mathcal{W}_q} = \left( \frac{\partial \mathcal{L}}{\partial \mathcal{R}} \odot \mathcal{R}_v \right) \cdot E_q^\top, \quad \nabla_{\mathcal{W}_v} \mathcal{L} = \frac{\partial \mathcal{L}}{\partial \mathcal{R}} \cdot \frac{\partial \mathcal{R}}{\partial \mathcal{R}_v} \cdot \frac{\partial \mathcal{R}_v}{\partial \mathcal{W}_v} = \left( \frac{\partial \mathcal{L}}{\partial \mathcal{R}} \odot \mathcal{R}_q \right) \cdot E_v^\top, \tag{3}$$

where $E_q$ and $E_v$ is the question embeddings and image embeddings, separately. The corresponding Frobenius norms are:

$$\| \frac{\partial \mathcal{L}}{\partial \mathcal{W}_q} \|_F = \| \frac{\partial \mathcal{L}}{\partial \mathcal{R}} \odot \mathcal{R}_v \|_2 \cdot \| E_q \|_2, \quad \| \frac{\partial \mathcal{L}}{\partial \mathcal{W}_v} \|_F = \| \frac{\partial \mathcal{L}}{\partial \mathcal{R}} \odot \mathcal{R}_q \|_2 \cdot \| E_v \|_2. \tag{4}$$

After experimental exploration, we found that $\| E_q \|_2 < \| E_v \|_2$ and $\| \frac{\partial \mathcal{L}}{\partial \mathcal{R}} \odot \mathcal{R}_v \|_2 \gg \| \frac{\partial \mathcal{L}}{\partial \mathcal{R}} \odot \mathcal{R}_q \|_2$, which yields $\| \frac{\partial \mathcal{L}}{\partial \mathcal{W}_q} \|_F > \| \frac{\partial \mathcal{L}}{\partial \mathcal{W}_v} \|_F$ (see Fig. 1(b)). In other words, **modalities gradient optimization deviation arises from inherent data heterogeneity.** Specifically, question tokens are encoded into 300-dimensional GloVe embeddings [31] and aggregated via a single-layer GRU [11], whereas visual inputs are represented by 36 object vectors of dimension 2048 extracted by a Faster R-CNN [33]. Consequently, $\| E_q \|_2 < \| E_v \|_2$ and $\| \frac{\partial \mathcal{L}}{\partial \mathcal{R}} \odot \mathcal{R}_v \|_2 \gg \| \frac{\partial \mathcal{L}}{\partial \mathcal{R}} \odot \mathcal{R}_q \|_2$. **This difference in the inherent properties of the data is the culprit behind the imbalance in modal optimization.**

Crucially, gradient deviation can not directly explain language bias, since the final predictions depend on fused representations $\mathcal{R}$ and classifier weights $\mathcal{W}$. To uncover the true bias formation, we next analyze (1) the proportion of unimodal components in $\mathcal{R}$ and (2) the directional deviation of $\mathcal{W}$.

### 2.2   Feature Fusion Bias and Classifier Deviation

**Fusion feature component deviation.**   In the previous subsection, we identified the modality gradient optimization deviation. Then we naturally ponder the question: **Which modality predominates the fusion feature?** Since the loss is essentially computed using fusion feature, this composition deviation determines the model's reliance on visual versus textual cues at prediction time [26].

Inspired by [19], given $\mathcal{R}$, $\mathcal{R}_q$, and $\mathcal{R}_v$, we perform Singular Value Decomposition (SVD) and obtain the right-singular unitary matrices $V_\mathcal{R}$, $V_{\mathcal{R}_q}$, and $V_{\mathcal{R}_v}$. We assess the similarities of the subspace

spanned by the top-$i$ singular vectors in $V_\mathcal{R}$ and that of the top-$j$ singular vectors of $V_{\mathcal{R}_q}$ and $V_{\mathcal{R}_v}$, respectively. We compute the normalized subspace similarity based on the Grassmann distance as:

$$\psi(V_\mathcal{R}, V_{\mathcal{R}_q/\mathcal{R}_v}, i, j) = \frac{\|V_\mathcal{R}^{:,:i\top} V_{\mathcal{R}_q/\mathcal{R}_v}^{:,:j\top}\|_F^2}{\min(i,j)} \in [0,1]. \tag{5}$$

Here, $\psi(\cdot)$ ranges from 0 to 1, where 1 indicates a complete overlap of subspaces and 0 signifies total separation. $V_\mathcal{R}^{:,:i}$ and $V_{\mathcal{R}_q/\mathcal{R}_v}^{:,:j}$ represents the top-$i$ and top-$j$ column vectors of $V_\mathcal{R}$ and $V_{\mathcal{R}_q/\mathcal{R}_v}$, respectively. As shown in Fig. 1(c), we observed an important phenomenon: **At the start of training, the question-subspace similarity is substantially lower than that of the image subspace, but by the mid- and late-training stages it exceeds the image-subspace similarity.** These findings reveal that the fused features are disproportionately governed by the question features, with visual features playing a subordinate role. **Such composition deviation causes the model to assign uneven importance to modality-specific information when learning from fused features.**

**Classifier weight direction deviation.** In general, the model stores key information about the classification task captured from the training data in its final classification weights. We perform SVD on $\mathcal{W}$ to extract corresponding singular values $\Sigma$. As depicted in Fig. 1(d), the singular values appear significantly different. The top singular value greatly exceeds the remaining values, and the bottom singular value is almost 0. This implies that the classifier predominantly captures feature correlations along a primary axis, neglecting orthogonal (secondary) directions that may encode complementary information. To elucidate which features are emphasized, we consider the gradient of $\mathcal{L}_{\mathrm{CE}}$ for $\mathcal{W}$:

$$\nabla_\mathcal{W} \mathcal{L}_{\mathrm{CE}} = (p - a)\mathcal{R}, \tag{6}$$

where $p = \frac{\exp(f_i)}{\sum_{j=1}^{|A|} \exp(f_j)}$. Here, the update for the correct class aligns exactly with the fused representation $\mathcal{R}$. Since $\mathcal{R}$ is dominated by the question component (see above), the weight updates encode question-driven priors, while visual-semantic cues in secondary directions are largely ignored.

## 3 Why Does the Margin Mechanism Work?

**Normalized *CE* loss for hypersphere embedding.** To optimize the instance representation space, prior works [27, 16, 4, 46, 47] project features onto a unit hypersphere by $L2-$ normalizing classifier weights $\mathcal{W}$ and joint representations $\mathcal{R}$. In light of this, the posterior probability is determined by the angle $\theta_i$ between $\mathcal{W}_i$ and $\mathcal{R}_i$, and the answer feature space is converted from the Euclidean space to the angular space. The logit $f_i$ for each representation $\mathcal{R}_i$ is redefined as:

$$f_i = \mathcal{W}_i^\top \mathcal{R}_i = \|\mathcal{W}_i\|\|\mathcal{R}_i\| s \cos\theta_i = s \cos\theta_i, \tag{7}$$

where $\|\mathcal{W}_i\| = 1, \|\mathcal{R}_i\| = 1$, $s$ is a scaling factor for more stable computation. The bias term is viewed as zero for simplicity. Thus, the joint representations $\mathcal{R}$ are distributed on a hypersphere with a radius $s$. The standard *CE* loss is transformed into a normalized *CE* loss:

$$\mathcal{L}_{\mathrm{NCE}} = \sum_{i=1}^{|\mathcal{A}|} -a_i \log \frac{\exp(s \cos\theta_i)}{\sum_{j=1}^{|\mathcal{A}|} \exp(s \cos\theta_j)}. \tag{8}$$

Rigorously, we theoretically analyze the rationale for spherical space learning and demonstrate its advantages in the supplementary materials. Recent studies [4, 47] have focused on optimizing the instance spacing in inverse cosine space by adding a margin $m$ to the clamp angle $\theta$:

$$\mathcal{L}_{\mathrm{MARGIN}} = \sum_{i=1}^{|\mathcal{A}|} -a_i \log \frac{\exp(s \cos(\theta_i + m_i))}{\sum_{j=1}^{|\mathcal{A}|} \exp(s \cos(\theta_j + m_j))}. \tag{9}$$

### 3.1 Modality Gradient Optimization Balance

We analyze how the margin term $m$ modulates encoder gradients by comparing the gradients of $\mathcal{L}_{\mathrm{CE}}$ and $\mathcal{L}_{\mathrm{MARGIN}}$ concerning the fused feature $\mathcal{R}$:

$$\nabla_\mathcal{R} \mathcal{L}_{\mathrm{CE}} = \mathcal{W}^\top(p - a), \tag{10}$$

$$\nabla_\mathcal{R} \mathcal{L}_{\mathrm{MARGIN}} = s\mathcal{W}^\top((p' - a) \odot \mathcal{C}), \text{ where } \mathcal{C} = \cos m + \cot\theta \cdot \sin m, \tag{11}$$

where $p' = \frac{\exp(f'_i)}{\sum_{j=1}^{|A|}\exp(f'_j)}$, $f'_i = s\cos(\theta_i + m_i)$. The margin mechanism thus (1) introduces the scale $s$ to amplify gradient magnitude, (2) perturbs the predicted probabilities $p'$ to reweight class-wise errors, and (3) applies an adaptive coefficient $\mathcal{C}$ along each logit dimension. As each margin $m_i$ and angle $\theta_i$ are independently specified, $\mathcal{C}$ differentially scales the alignment of $\nabla_{\mathcal{R}}$ with the question subvector $\mathcal{R}_q$ versus the visual subvector $\mathcal{R}_v$. Consequently, the margin term rebalances modality-specific gradient contributions and reduces the gradient deviation (see Fig. 1(b)).

## 3.2 Fusion Feature Component Uniformity

Fig. 1(c) illustrates that the question-modality contribution within the fused representation grows sharply and ultimately surpasses the image-modality contribution, resulting from disproportionate gradient updates favoring the question stream. In the previous subsection, we demonstrate that the margin mechanism counteracts this imbalance by harmonizing modality gradient magnitudes, thereby producing a more uniform multimodal feature composition.

## 3.3 Classifier Weight Direction Equalization

The margin $m$ mitigates directional bias in $\mathcal{W}$ by promoting a more uniform singular spectrum. The gradient of $\nabla_{\mathcal{W}}\mathcal{L}_{\text{MARGIN}}$ with respect to $\mathcal{W}$ is:

$$\nabla_{\mathcal{W}}\mathcal{L}_{\text{MARGIN}} = s((p' - a) \odot \mathcal{C})\mathcal{R}. \tag{12}$$

The first term of $\mathcal{C}$, $\cos m \in [-1, 1]$, uniformly scales updates along $\mathcal{R}$, attenuating the dominant (question-driven) component more strongly because of its larger magnitude. We then consider the second term of $\mathcal{C}$, $\cot\theta \cdot \sin m$. Defining the unit vector $\hat{\mathcal{W}}_i = \frac{\mathcal{W}}{\|\mathcal{W}_i\|}$, and $\mathcal{U}$ as a unit vector orthogonal to $\hat{\mathcal{W}}$, we geometrically decompose $\mathcal{R}$ into $\hat{\mathcal{W}}$ and $\mathcal{U}$:

$$\mathcal{R} = (\hat{\mathcal{W}}^\top\mathcal{R}\hat{\mathcal{W}}) + (\mathcal{R} - (\hat{\mathcal{W}}^\top\mathcal{R}\hat{\mathcal{W}})) = \cos\theta \cdot \hat{\mathcal{W}} + \sin\theta \cdot \mathcal{U}. \tag{13}$$

The second term of $\mathcal{C}$, $\cot\theta \cdot \sin m$ yields:

$$\cot\theta \cdot \sin m \cdot \mathcal{R} = \frac{\cos^2\theta}{\sin\theta} \cdot \sin m \cdot \hat{\mathcal{W}} + \cos\theta \cdot \sin m \cdot \mathcal{U}. \tag{14}$$

Here, $\frac{\cos^2\theta}{\sin\theta} \cdot \sin m \cdot \hat{\mathcal{W}}$ balances the principal component adaptively, while $\cos\theta \cdot \sin m \cdot \mathcal{U}$ injects a corrective push into the orthogonal subspace. As a result, primary singular values decrease and secondary singular values increase, equalizing the classifier's learned directions and enhancing its ability to capture complementary cues. We provide a more rigorous theoretical proof in the supplementary materials.

# 4 Methodology

## 4.1 Multi-Grained Adaptive Margins

Inspired by margin learning [16, 4], our MAM mechanism aims to address the challenge of chaotic class boundaries posed by imbalanced data. By considering answer frequency and evaluating instance difficulty from coarse-grained and fine-grained perspectives, MAM enhances intra-class compactness and inter-class separation, thus refining a discriminative and robust feature space. Specifically, MAM integrates three components: frequency-aware, confidence-aware, and difficulty-aware margins.

**Frequency-aware margins.** As mentioned in [16], imposing larger margin penalties on minority classes is crucial for driving their representations closer to the respective class centers. Conversely, majority classes, which naturally have a robust representation, benefit from smaller margin penalties. Similar to [16, 4], the frequency-aware margins are defined as:

$$\hat{m}_i^{qt} = \frac{n_i^{qt} + \epsilon}{\sum_{j=1}^{|\mathcal{A}|} n_j^{qt} + \epsilon}, \tag{15}$$

where $\hat{m}_i^{qt}$ is the frequency of answer $a_i$ with question type $qt$. $n_i$ is the occurrence of answer $a_i$ with $qt$. $\epsilon$ is a hyperparameter for avoiding computational overflow. Elasticface [5] demonstrates that

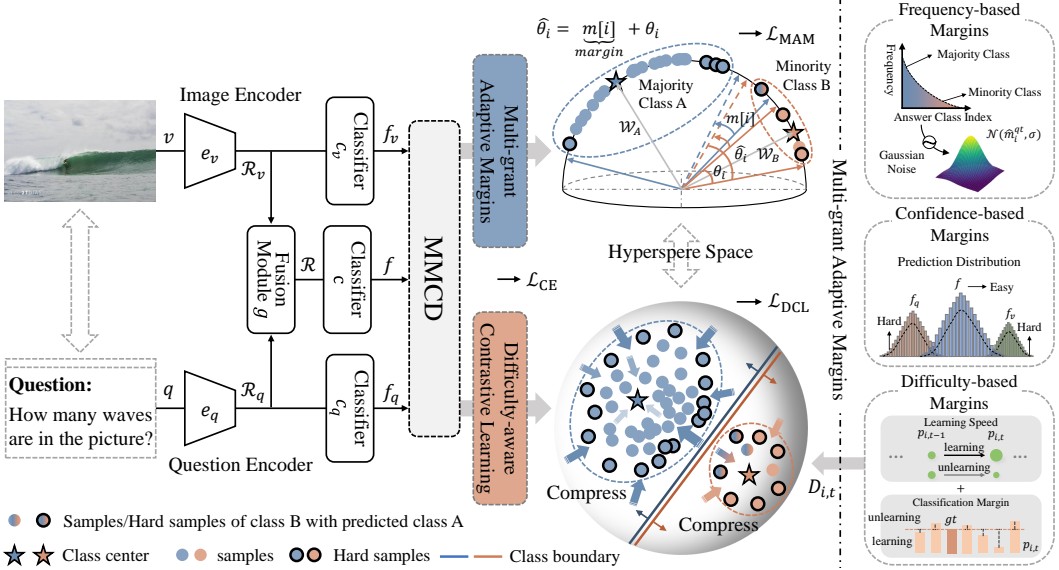

Figure 2: Illustration of our MMCD for combating language bias. *Up: Multi-Grained Adaptive Margins* rectify instance to reshape robust class boundaries. *Bottom: Difficult-aware Contrastive Learning* improves intra-class compactness and inter-class separation, carving discriminative feature space. *Right: Multi-Grained Adaptive Margins* are driven by frequency and instance difficulty.

fixed margins fail to adapt to the dynamic inter- and intra-class variances in real-world data, thereby impairing the model's discriminability and generalization. Following [4], we incorporate random Gaussian noise into frequency-aware margins $m_{freq}$:

$$m_i^{qt} = \mathcal{N}(\hat{m}_i^{qt}, \sigma), \tag{16}$$

where $\mathcal{N}$ is the Gaussian distribution with standard deviation $\sigma$, and $\sigma$ is a hyperparameter.

**Confidence-aware margins.** As mentioned above, sample difficulty significantly affects the discriminative decision margin and class separability. A simple yet effective measure of difficulty is the prediction logits. We incorporate auxiliary branches dedicated solely to the question and image modalities. These branches promote multimodal integration by deliberately introducing controlled modality bias. This strategy not only boosts ID performance but also prevents excessive bias correction.

Specifically, we introduce a question-only classifier $c_q$ and an image-only classifier $c_v$. The corresponding logits $f_q$ and $f_v$ are:

$$f_q(q) = c_q(e_q(q)), \quad f_v(v) = c_v(e_v(v)). \tag{17}$$

Recognizing the challenges of imbalanced multimodal learning [39, 40], inspired by [30], we leverage the posterior distributions $s_q$ and $s_v$ as the weights for unimodal logits $f_q$ and $f_v$, respectively.

$$s_q = \text{softmax}(\mathcal{W}_q \cdot e_q(q) + \frac{b}{2})[gt], \quad s_v = \text{softmax}(\mathcal{W}_v \cdot e_v(v) + \frac{b}{2})[gt], \tag{18}$$

where $\mathcal{W}_q$ and $\mathcal{W}_v$ are the parameters of $c_q$ and $c_v$ respectively. $gt$ refers to the index of the ground truth class of the sample. We denote $s_f = f(v, q)$ as the weight applied to the multimodal logits $f$, and $\tau_1$ as the temperature, which is a hyperparameter. Based on these definitions, the weighted hybrid confidence $f_m$ and confidence-aware margins $m_{conf}$ are formulated as follows:

$$f_m = \frac{s_f \cdot f + s_q \cdot f_q + s_v \cdot f_v}{s_f + s_q + s_v}, \quad m_{conf} = \text{softmax}(f_m/\tau_1), \tag{19}$$

**Difficulty-aware margins.** Although logits can simply and intuitively reflect sample difficulty, their static nature limits the fine-grained mining of intrinsic sample difficulty and makes it difficult to modulate stubborn decision boundaries. Inspired by [44], we develop a fine-grained difficulty

model that evaluates instance difficulty from two perspectives: **a) Learning Rate:** akin to human learning, where easy samples are learned quickly, and **b) Classification Margins:** reflecting relative confidence, where smaller margins indicate closer proximity to the decision boundary. Specifically, given an instance representation $\mathcal{R}_i$, its difficulty $D_{i,t}$ after $t$ iterations is estimated as:

$$D_{i,t} = \alpha \cdot \underbrace{\frac{vu_{i,t} + c}{vl_{i,t} + c}}_{\text{learning speed}} + (1 - \alpha) \cdot \underbrace{\frac{mu_{i,t} + c}{ml_{i,t} + c}}_{\text{classification margins}}, \tag{20}$$

where $vu_{i,t}$ and $mu_{i,t}$ denote the prediction variation and marginal gap on the unlearning direction after $t$ iterations respectively, $vl_{i,t}$ and $ml_{i,t}$ denote the prediction variation and marginal gap on the learning direction separately. $\alpha$ is a hyperparameter to balance the contribution of learning speed and classification margins. $c$ is a prior parameter to control the sensitivity of $D_{i,t}$ for the variation of predictions and prevent division by zero. A larger $D_{i,t}$ indicates that the instance is difficult to learn.

We apply Jensen-Shannon (JS) divergence to quantify learning speed. Specifically, we denote $p_{i,t}$ as the prediction distribution of instance $\mathcal{R}_i$ at $t$ iteration and $p_{i,t-1}$ at $t-1$ iteration. The distance $v_{i,t}$ between $p_{i,t-1}$ and $p_{i,t}$ is defined as:

$$v_{i,t} = \frac{1}{2}\text{KL}(p_{i,t-1}\|q_{i,t}) + \frac{1}{2}\text{KL}(p_{i,t}\|q_{i,t}), \tag{21}$$

where $q_{i,t} = (p_{i,t-1} + p_{i,t})/2$, KL is Kullback-Leibler (KL) divergence. We denote $p_{i,t}^j$ as the probability of class $j$ of $\mathcal{R}_i$ at the $t$ iteration. Obviously, $p_{i,t}^{gt} - p_{i,t-1}^{gt} < 0$ or $p_{i,t}^j - p_{i,t-1}^j > 0, j = 1\ldots C, j \neq gt$ indicates unlearning, and $p_{i,t}^{gt} - p_{i,t-1}^{gt} > 0$ or $p_{i,t}^j - p_{i,t-1}^j < 0, j = 1\ldots C, j \neq gt$ indicates learning, $C$ is the number of candidate answers. Therefore, $vu_{i,t}$ and $vl_{i,t}$ can be defined as:

$$\begin{cases} vu_{i,t} = \beta \cdot vu_{i,t-1} + (1 - \beta) \cdot vu'_{i,t}, \quad vl_{i,t} = \beta \cdot vl_{i,t-1} + (1 - \beta) \cdot vl'_{i,t}, \\[2mm] vu'_{i,t} = \min(p_{i,t}^{gt} - p_{i,t-1}^{gt}, 0)v_{i,t}[gt] + \sum_{j=1,j\neq gt}^{C} \max(p_{i,t}^j - p_{i,t-1}^j, 0)v_{i,t}[j], \\[2mm] vl'_{i,t} = \max(p_{i,t}^{gt} - p_{i,t-1}^{gt}, 0)v_{i,t}[gt] + \sum_{j=1,j\neq gt}^{C} \min(p_{i,t}^j - p_{i,t-1}^j, 0)v_{i,t}[j], \end{cases} \tag{22}$$

which satisfy that $v_{i,t} = vu'_{i,t} + vl'_{i,t}$. $[\cdot]$ denotes the index operator. $p_{i,0} = 1/C$ for all instances. $\beta$ is a hyperparameter used to weight historical and real-time information, thus preserving historical trends while being sensitive to short-term changes. Furthermore, we innovatively quantify instance difficulty through classification margins. Specifically, the classification margins $m_{i,t}$ is defined as:

$$m_{i,t} = |p_{i,t}^{gt} - p_{i,t}^j|, \quad j = 1\ldots C, j \neq gt, \tag{23}$$

where $|\cdot|$ denotes the absolute value operation. Apparently, $p_{i,t}^{gt} - p_{i,t}^j > 0$ indicates learning, $p_{i,t}^{gt} - p_{i,t}^j < 0$ indicates unlearning. Therefore, $mu_{i,t}$ and $ml_{i,t}$ can be defined as:

$$\begin{cases} mu_{i,t} = \beta \cdot mu_{i,t-1} + (1 - \beta) \cdot mu'_{i,t}, \quad ml_{i,t} = \beta \cdot ml_{i,t-1} + (1 - \beta) \cdot ml'_{i,t}, \\[2mm] mu'_{i,t} = \log(\frac{1}{|\Psi|} \sum_{j\in\Psi} \exp(m_{i,t}^j)), \qquad ml'_{i,t} = \log(\frac{1}{|\Omega|} \sum_{j\in\Omega} \exp(m_{i,t}^j)), \end{cases} \tag{24}$$

where $\Psi$ is the set of class $j$ satisfying $p_{i,t}^j - p_{i,t}^{gt} > 0$ and $\Omega$ is the set of class $j$ satisfying $p_{i,t}^j - p_{i,t}^{gt} < 0$. The difficulty-aware margins $m_{diff}$ are defined as:

$$m_{diff} = 1 - \text{softmax}(D_{i,t}), \tag{25}$$

**Margin loss formulation.** Ultimately, the various margin terms are aggregated in a cohesive manner to form the multi-grained adaptive margins $m_{\text{MAM}}$:

$$\begin{cases} m_{\text{MAM}} = m_{freq}, \\ m_{\text{MAM}}[gt] = (1 - \lambda_1) \cdot m_{\text{MAM}}[gt] + \lambda_1 \cdot m_{conf}[gt], \\ m_{\text{MAM}}[gt] = (1 - \lambda_2) \cdot m_{\text{MAM}}[gt] + \lambda_2 \cdot m_{diff}, \quad \text{epoch} \geq w, \\ m_{\text{MAM}} = 1 - m_{\text{MAM}}, \end{cases} \tag{26}$$

Table 1: Accuracy comparisons with other methods on the VQA-CP v2 and VQA-CP v1 datasets.

| Datasets | | VQA-CP v2 | | | | VQA-CP v1 | | | |
|---|---|---|---|---|---|---|---|---|---|
| Methods | | All | Y/N | Num | Others | All | Y/N | Num | Others |
| UpDn [2] | CVPR'18 | 39.74 | 42.27 | 11.93 | 46.05 | 37.96 | 42.79 | 12.41 | 42.53 |
| RUBi [6] | NeurIPS'19 | 47.11 | 68.65 | 20.28 | 43.18 | - | - | - | - |
| LMH [12] | EMNLP'19 | 52.15 | 70.29 | 44.10 | 44.86 | 55.73 | 78.59 | 24.68 | 45.47 |
| GGE-iter [17] | ICCV'21 | 57.12 | 87.35 | 26.16 | 49.77 | 59.82 | 85.52 | 28.93 | 46.67 |
| AdaVQA [16] | IJCAI'21 | 54.02 | 70.83 | 49.00 | 46.29 | 61.20 | **91.17** | 41.34 | 39.38 |
| COB [20] | WACV'23 | 57.53 | 88.36 | 28.81 | 49.27 | 60.98 | 87.41 | 32.02 | 46.34 |
| GENB [10] | CVPR'23 | 59.15 | 88.03 | 40.05 | 49.25 | 62.74 | 86.18 | 43.85 | **47.03** |
| GGD [18] | TPAMI'23 | 59.37 | 88.23 | 38.11 | 49.82 | - | - | - | - |
| CVIV [29] | TMM'24 | 60.08 | 88.85 | 40.77 | **50.30** | - | - | - | - |
| PWVQA [36] | TMM'24 | 59.06 | 88.26 | 52.89 | 45.45 | - | - | - | - |
| MMCD | Ours | **61.34** | **88.93** | **55.68** | 48.44 | **63.62** | 90.72 | **52.67** | 41.08 |

Table 2: Performance of our approach with different network architectures

| Methods | All | Y/N | Num | Other | Increased ↑ |
|---|---|---|---|---|---|
| SAN | 26.88 | 35.34 | 11.34 | 24.70 | |
| SAN+MMCD | **60.12** | **86.97** | **54.62** | **47.56** | 33.24 |
| S-MRL | 38.46 | 42.85 | 12.81 | 43.20 | |
| S-MRL+MMCD | **60.69** | **88.40** | **55.44** | **47.61** | 22.23 |
| LXMERT | 48.66 | 47.49 | 22.24 | **56.52** | |
| LXMERT+MMCD | **66.95** | **91.79** | **63.28** | 54.39 | 18.29 |

where $w$ denotes a specific epoch that marks the end of the warm-up stage. The $m_{\mathrm{MAM}}$ are added to the angle $\theta_i$ as a margin penalty:

$$\mathcal{L}_{\mathrm{MAM}} = \sum_{i=1}^{|\mathcal{A}|} -a_i \log \frac{\exp(s \cos(\theta_i + m_{\mathrm{MAM}}))}{\sum_{j=1}^{|\mathcal{A}|} \exp(s \cos(\theta_j + m_{\mathrm{MAM}}))}. \tag{27}$$

## 4.2 Difficulty-aware Contrastive Learning

We further propose the DCL mechanism that integrates our instance difficulty model into a supervised contrastive paradigm [23], which dynamically emphasizes hard samples by difficulty-adaptive weighting, effectively enhancing intra-class compactness and inter-class separation to form a discriminative feature space. Specifically, we consider a mini-batch $\mathcal{B} = \{(x_1, a_1), (x_2, a_2), \ldots, (x_{|\mathcal{B}|}, a_{|\mathcal{B}|})\}$ of $L_2$−normalized joint representations $\mathcal{R}$ and corresponding answers $a_i$. For each anchor feature $x_j$ with difficulty $D_{j,t}$ at $t$ iteration, the positive set $P_j = \{i \in \mathcal{B} \mid a_i = a_j, i \neq j\}$ contains indices of all non-anchor samples with identical answers, and the negative set $N_j = \{i \in \mathcal{B} \mid a_i \neq a_j\}$ includes indices of samples with different answers. The DCL loss is formulated as:

$$\mathcal{L}_{\mathrm{DCL}} = \sum_{j \in \mathcal{B}} \frac{-1}{|P_j|} \sum_{p \in P_j} \log \frac{\exp(D_{p,t}) \exp(x_j^\top x_p / \tau_2)}{\sum_{n \in N_j} \exp(D_{n,t}) \exp(x_j^\top x_n / \tau_2)}, \tag{28}$$

where temperature $\tau_2$ is set to 1.0.

## 4.3 Training and Optimization

Based on the above analyses, the comprehensive training objective of the proposed MMCD approach encompasses a combination of various loss functions, i.e.,

$$\mathcal{L}_{\mathrm{TOTAL}} = \begin{cases} \mathcal{L}_{\mathrm{CE}} + \mathcal{L}_{\mathrm{MAM}} + \lambda_3 \mathcal{L}_{\mathrm{SupCon}}, & \text{epoch} < w, \\ \mathcal{L}_{\mathrm{CE}} + \mathcal{L}_{\mathrm{MAM}} + \lambda_3 \mathcal{L}_{\mathrm{DCL}}, & \text{epoch} \geq w, \end{cases} \tag{29}$$

where $\mathcal{L}_{\mathrm{SupCon}}$ is the standard supervised contrastive loss.

Table 3: Ablation experiments for different modules of the MMCD model on VQA-CP v2.

| Methods | Frequency-aware Margins | Confidence-aware Margins | Difficulty-aware Margins | DCL | All |
|---|---|---|---|---|---|
| Baseline | | | | | 39.74 |
| Variant-I | ✓ | | | | 59.44 |
| Variant-II | ✓ | ✓ | | | 59.74 |
| Variant-III | ✓ | ✓ | ✓ | | 61.09 |
| Variant-IV | | | | ✓ | 41.09 |
| MMCD (Ours) | ✓ | ✓ | ✓ | ✓ | **61.34** |

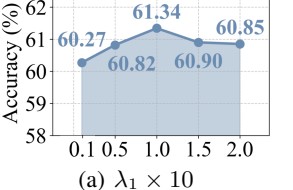
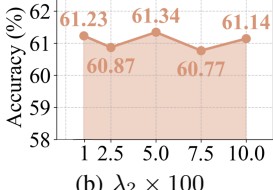
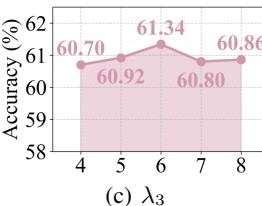

Figure 3: Comparison of Accuracy on the VQA-CP v2 dataset with different parameter configurations.

# 5 Experiments

## 5.1 Datasets & Implementation Details

We select various OOD benchmarks to assess the robustness of models against real-world biases, such as VQA-CP v2, VQA-CP v1 [1]. All experiments adopt the standard evaluation metric [3]. Further details on the experimental setup and implementation can be found in the supplementary materials.

## 5.2 Comparisons with State-of-the-Arts

As shown in Table 1, we report both the overall accuracy and the per-category performance across question types, including "yes/no", "number", and "other". Compared to the second-best method, MMCD achieves gains of 1.26% and 0.72% in overall accuracy on the VQA-CP v2 and VQA-CP v1 datasets, respectively. Notably, MMCD achieves state-of-the-art performance on the "yes/no" and "number" categories, which are typically more susceptible to language priors. In particular, MMCD yields a substantial 2.79% improvement in the "number" category on VQA-CP v2, demonstrating its strong ability to mitigate language bias and enhance reasoning over numerical questions.

## 5.3 Extensive Experiments with Different Architectures

We further evaluate the generalizability and robustness of MMCD across additional architectures, including SAN [43], S-MRL [6], and LXMERT [34]. As shown in Table 2, the MMCD approach consistently outperforms the corresponding baselines, demonstrating strong adaptability and model-agnostic performance across diverse network designs. In particular, applying MMCD to LXMERT, a widely adopted vision-language pre-trained model commonly used in various multimodal downstream tasks, yields a notable 18.29% performance improvement, further highlighting its effectiveness in enhancing a broad range of model families.

## 5.4 Ablation Studies.

To assess the impact of each component in the proposed MMCD method, we perform a series of ablation experiments with various variations. The comparison results of the ablation study are shown in Table 3. Specifically, **Variant-I** outperforms baseline by 18.02%. The substantial performance enhancement demonstrates its critical function in mitigating language biases resulting from class imbalance by utilizing prior information. Furthermore, by incorporating confidence-aware margins, **Variant-II** achieves 0.3% performance gains compared to **Variant-I**, suggesting that the simple multimodal logits strategy effectively utilizes the inherent sample complexity and results in the

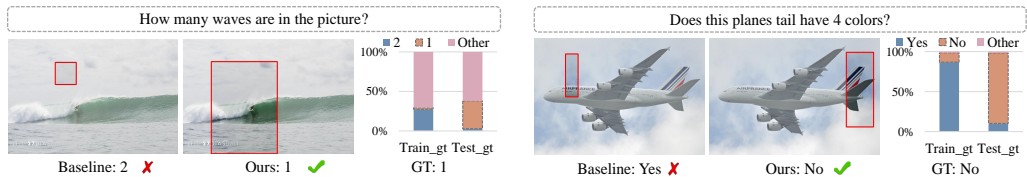

Figure 4: Visualization results of MMCD in robust reasoning and bias mitigation.

development of robust and discriminative feature spaces. With the integration of Difficulty-aware margins, **Variant-III** combines into a complete multi-grained adaptive margins mechanism, making a significant contribution to shaping a more structured and organized spherical representation space. Subsequently, **Variant-IV** emphasizes the pivotal role of the DCL mechanism in enhancing intra-class compactness and inter-class separation. Eventually, our full **MMCD** model achieves the best performance, demonstrating the effectiveness and availability of the designed components. More ablation analysis can be found in the supplementary materials.

### 5.5 Parameter Analysis.

As shown in Fig. 3, we systematically evaluate MMCD under a comprehensive range of hyperparameter settings. Our study focuses on three key hyperparameters: $\lambda_1$ and $\lambda_2$ in Eqn. (26), and $\lambda_3$ in Eqn. (29). Across all combinations, accuracy fluctuates by no more than 1.07%. Such minimal variance highlights MMCD's robustness to hyperparameter selection, significantly reducing the need for exhaustive tuning. Moreover, this stability suggests that MMCD can be readily transferred to new VQA benchmarks or application domains without extensive reconfiguration. Overall, the insensitivity to parameter settings not only simplifies deployment but also confirms MMCD's strong generalization capability in mitigating language bias. More parameter analysis can be found in the supplementary materials.

### 5.6 Qualitative Analysis.

As shown in Fig. 4, the MMCD method not only accurately localizes the correct area but also exhibits exceptional performance in removing bias. By enforcing a well-structured feature space, our approach facilitates the learning of highly discriminative features and the extraction of high-quality multimodal information. This structured representation enhances the model's ability to identify and leverage key visual cues, which is critical for robust performance in VQA tasks.

## 6 Conclusion

In this paper, we investigated the origin of language bias in VQA and elucidated why margin-based mechanisms effectively mitigate it. Empirical evidence shows that multimodal data heterogeneity induces gradient optimization imbalances, leading to biased feature fusion and classifier weight deviations. We provide theoretical support from both gradient and spectral perspectives, demonstrating how margin-based objectives counteract these effects. Building on these insights, we propose MMCD, an adaptive multi-margin framework that incorporates sample frequency and difficulty to reshape decision boundaries and enhance feature discrimination via difficulty-aware contrastive learning. Extensive experiments confirm the superior robustness of MMCD, with potential implications for broader challenges such as shortcut learning, long-tail recognition, and class imbalance.

## Acknowledgments

This work was supported in part by the Guangdong Basic and Applied Basic Research Foundation (Nos. 2025A1515010225, 2025A1515060001), in part by the National Natural Science Foundation of China (No. 62302172), and in part by the JSPS KAKENHI (No. JP25K21207) and JST CREST (No. JPMJCR22M2).

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
