# OpenReview forum: "Language‑Bias‑Resilient Visual Question Answering via Adaptive Multi‑Margin Collaborative Debiasing"
_NeurIPS.cc/2025/Conference — NeurIPS 2025 poster_

### Official Review · Reviewer_qcZ8 · 2025-06-20

**Clarity:** 3
**Significance:** 3
**Originality:** 4
**Rating:** 5
**Confidence:** 4

**Summary:**

This paper tackles performance degradation in Visual Question Answering (VQA) caused by language bias. Through theoretical and empirical analysis, the authors identify modality-specific gradient optimization deviation, which results in feature fusion skewness and classifier weight imbalance. To mitigate this, they propose MMCD (Multi-Margin Collaborative Debiasing), a framework that reshapes decision boundaries under biased training via three adaptive angular margins (frequency, confidence, and difficulty-aware) and difficulty-aware contrastive learning mechanism. Experiments on multiple benchmarks demonstrate the effectiveness of MMCD.

**Questions:**

Pls refer to Strengths and weaknesses.

**Ethical Concerns:**

["NO or VERY MINOR ethics concerns only"]

**Final Justification:**

I have read the authors’ rebuttal and the other reviewers’ comments, the authors have addressed most of my concerns, and thus I consist of my original score.

**Limitations:**

yes

**Paper Formatting Concerns:**

N.A.

**Quality:**

4

**Strengths And Weaknesses:**

Strengths:
S1: Comprehensive and in-depth analysis: Through a combination of empirical and theoretical analyses, the authors systematically explore the formation mechanism of language bias from modal gradient optimization deviation, feature fusion deviation, and classifier weight direction deviation. Besides, the authors also provide theoretical support from the gradient and spectral perspectives as to why the margin-based mechanism can mitigate the bias, and the analysis is thorough.
S2: The proposed MMCD framework innovatively integrates frequency-aware, confidence-aware, and difficulty-aware angular margins, as well as a dynamic, difficulty-aware contrast learning mechanism.Experiments show that the framework outperforms the existing baseline in several challenging VQA benchmarks, effectively countering language bias.
S3: The writing is well-organized and easy to follow.

	Weaknesses:
W1: MMCD introduces multiple branches (e.g., unimodal classifiers) and a difficulty estimation mechanism that bring additional calculation expense, and it is recommended to add an experimental analysis of the computational efficiency (inference time, runtime memory usage) to assess the feasibility of real-world deployment.
W2: MAM mechanism relies on priors of question types and answer distributions, is it generalizable under different data distributions? It is recommended to conduct experiments on GQA/ScienceQA to verify the generalization performance of the method.
W3: With the introduction of additional modules (FAM, CAM, DAM, DCL) in MMCD, several hyperparameters need to be tuned during the training process, so is there any difficulty in selecting ,combining and optimizing the hyperparameters with different network architectures and datasets?

---

> ### Author Rebuttal · Authors · 2025-07-31
>
> Thanks for your perceptive comments.
>
> **Q1: Clarification of Computational Cost.**
>
> ***All experiments adhere to standard VQA protocols while incurring only minimal computational overhead.*** During training, our MAM and DCL function solely as loss-level regularizers—no extra branches or contrastive modules are introduced. At inference, the model executes the identical forward pass as the baseline, so latency, FLOPs, and memory usage remain unchanged. As Table 1 shows, training FLOPs increase marginally from 0.18 G to 0.19 G, and the parameter count from 35.18 M to 35.9 M. **Compared to existing baselines, MMCD delivers significant robustness gains with negligible inference-time overhead and a modest training-time cost.**
>
> Table 1: Comparasion of FLOPs and Parameters on VQA-CP v2.
> | Methods | Ref. | FLOPs(G) | Parameters(M) |
> | -------- | -------- | -------- | -------- |
> | UpDn | CVPR'18 | 0.18 | 12.94 |
> | RMLVQA | CVPR'23 | 0.17 | 18.88 |
> | GGD | TPAMI'23 | 0.19 | 33.98 |
> | COB | WACV'23 | 0.16 | 35.18 |
> | PHOH | AAAI'25 | 0.31 | 18.88 |
> | **MMCD** | Ours | 0.19 | 35.90 |
>
> **Q2: Generalization Performance of the Method.**
>
> To further assess generalization, we applied MMCD to the GQA dataset, which emphasizes compositional reasoning and visual grounding. **As shown in Table 2, MMCD achieves an overall accuracy of 75.1%, exceeding the runner‐up by 14.8%. This substantial gain demonstrates that our adaptive multi‐grained margins effectively transfer to tasks requiring intricate reasoning under diverse distributions.**
>
> Importantly, our experimental suite spans both language‐bias benchmarks (VQA-CP v2, where prior answer statistics dominate) and multimodal‐bias benchmarks (VQA-CE). In each scenario, MMCD attains state‐of‐the‐art results, illustrating its capacity to harmonize modality contributions and mitigate spurious correlations. **These findings confirm that MMCD is not tailored to a single bias type or dataset but provides a universal framework for robust VQA across changing domains and bias patterns.**
>
> Table 2: Accuracy comparisons with other methods on GQA.
> | Methods | Ref. | Accuracy |
> | -------- | -------- | -------- |
> | UpDn     | CVPR'18 | 49.7 |
> | COB | WACV'23 | 42.1 |
> | MSCD | MM'24 | 60.3 |
> | **MMCD** | Ours | **75.1** |
>
> **Q3: Parameter Sensitivity.**
>
> **Table 3 reports MMCD’s performance across a comprehensive hyperparameter grid, where accuracy varies by at most 1.07%.** This negligible fluctuation demonstrates MMCD’s strong resilience to hyperparameter settings, greatly reducing the need for exhaustive tuning. Furthermore, such stability implies MMCD can be deployed on new VQA benchmarks or specialized domains with minimal reconfiguration. **In summary, MMCD’s parameter insensitivity not only streamlines practical deployment but also confirms its robust generalization for mitigating multimodal bias.**
>
> Table 3: Comparison of Accuracy on the VQA-CP v2 dataset with different parameter configurations.
> |λ_1| 0.01 | 0.05 | 0.1 | 0.15 | 0.2 |
> | -------- | -------- | -------- | -------- | -------- | -------- |
> |Accuracy(%)| 60.27 | 60.82 | **61.34** | 60.90 | 60.85 |
>
> |λ_2| 0.01 | 0.025 | 0.05 | 0.075 | 0.1 |
> | -------- | -------- | -------- | -------- | -------- | -------- |
> |Accuracy(%)| 61.23 | 60.87 | **61.34** | 60.77 | 61.14 |
>
> |α| 0.4 | 0.5 | 0.6 | 0.7 | 0.8 |
> | -------- | -------- | -------- | -------- | -------- | -------- |
> |Accuracy(%)| 61.11 | 60.95 | **61.34** | 60.73 | 60.82 |
>
> Finally, **we hope this clarification strengthens the impact of our work and improves its evaluation.**

---

### Official Review · Reviewer_UEGk · 2025-07-02

**Clarity:** 2
**Significance:** 3
**Originality:** 3
**Rating:** 5
**Confidence:** 2

**Summary:**

The paper presents an empirical and theoretical investigation into why VQA models develop language biases. The authors find that skewed gradient norms between the visual and question encoders lead to question-dominated fused features and classifier weights. They also explain why margin-based methods can help alleviate this issue. Building on these insights, the authors propose a debiasing method, and experiments involving different architectures demonstrate improvements over existing approaches.

**Questions:**

Please see the questions raise in the Weaknesses section.

**Ethical Concerns:**

["NO or VERY MINOR ethics concerns only"]

**Final Justification:**

I am overall satisfied with the rebuttal and have adjusted my score accordingly.

**Limitations:**

The limitations are discussed in Appendix K.

**Quality:**

2

**Strengths And Weaknesses:**

## Strengths

1. The paper offers an in-depth exploration combining experiments and mathematical derivation to uncover how biases arise in VQA models.

2. Experiments demonstrate that the proposed method is consistently effective across different architectures and setups.


## Weaknesses

1. The Method section is difficult to follow. It would benefit from a high-level roadmap: a brief overview of the intuition and purpose behind each component before diving into the technical details.

2. The caption for Figure 1 is confusing. Perhaps the current (c) is redundant, and the two (d)’s should instead be labeled (c) and (d). Additionally, the caption is not very informative—it would be helpful to briefly describe the elements shown in each plot.

3. The ablation study on the three components of Multi-Grained Adaptive Margins is limited (it currently appears in the appendix but would be better placed in the main paper). Instead of adding each component individually to the baseline, it would be more informative to remove one component at a time from the full method to assess each component’s marginal contribution.

4. Can the authors clarify how many hyperparameters the method has in total? From Figures 3 and 6, it appears there are six, which would make tuning very challenging in practice. Moreover, the plots seem to show that performance is quite sensitive to even a single hyperparameter changing from its optimal value, let alone simultaneous variations in multiple parameters.


5. As a non-expert in this specific topic, I cannot fully evaluate whether the baselines, comparisons, and experimental setups cover the state of the art; I welcome perspectives from other reviewers.

---

> ### Author Rebuttal · Authors · 2025-07-31
>
> Thanks for your discerning comments.
>
> **Q1: Motivation of Technology.**
>
> Unfortunately, we are not allowed to provide a high-level roadmap to showcase the intuition and purpose behind each component. We provide the following overview. **Margin learning effectively mitigates the chaotic class‐boundary issues induced by data imbalance.** Our multi-grained adaptive margins comprise coarse-grained margins and fine-grained margins by considering answer frequency and instance difficulty, respectively. **The frequency-aware margins are driven by the inherent properties of imbalanced data.** Minority classes rely on larger margin penalties to push towards the center of the class, while the robust features of most classes benefit from smaller margin penalties. By utilizing prior data information, appropriate boundary penalties can be applied at a coarse-grained level. **The confidence-aware margins are based on the difficulty of the sample.** The difficulty of the sample significantly affects the formation of discriminative decision boundaries and strong class separability. Predictive logits can easily and effectively evaluate sample difficulty. **Considering that logits cannot finely mine inherent instance difficulty, the difficulty-aware margins further construct a fine-grained difficulty model to analyze sample difficulty and enhance reasoning ability and robustness.** Overall, these complementary margins dynamically sculpt clear and robust decision boundaries across multiple granularities.
>
> **Q2: Confusion about the Caption.**
>
> We have reviewed the caption for Figure 1 and corrected it as pointed out. The current (b) and (c) are duplicated. The first (d) should be labeled (c). The revised caption is now included in the manuscript. We provide a brief description as follows.
> (a) Comparison of gradient norms between modalities: in the baseline UpDn model, the question‐mode gradient norm exceeds that of the image mode, exacerbating language bias; MMCD balances these norms.
> (b) Evolution of gradient bias during training: UpDn maintains a higher question‐mode gradient than image‐mode gradient, whereas MMCD progressively reduces this discrepancy.
> (c) Modality–fusion feature similarity: UpDn shows a larger similarity for question features versus fusion features and a smaller one for image features; MMCD equalizes these similarities across modalities.
> (d) Classifier singular value distribution: UpDn’s top singular values dominate, indicating directional bias that captures only primary features.
>
> **Q3: More Ablation Study.**
>
> We conducted an ablation study by sequentially removing each component of the multi‐grained adaptive margin. As summarized in Table 1, omitting any single margin degrades overall accuracy, confirming their complementary roles. In particular, the frequency-aware margins have the greatest impact: its removal incurs a 7.26% drop in accuracy, underscoring the importance of leveraging class‐frequency priors to mitigate language bias. Furthermore, performance on “yes/no” and “other” questions remains largely unaffected, demonstrating MMCD’s consistent reasoning ability, whereas accuracy for “number” questions declines markedly, indicating heightened sensitivity to numerical responses.
>
> Table 1: Ablation experiments for different modules of the MMCD model on the VQA-CP v2 dataset. We remove each component one by one from the complete method to evaluate the corresponding marginal contribution.
> | Methods | All | Y/N | Num | Others |
> | -------- | -------- | -------- | -------- | -------- |
> | MMCD | **61.34**| 88.93 | **55.68** | 48.44 |
> | -Frequency-aware Margins | 54.08 | 86.43 | 18.88 | 46.79 |
> | -Confidence-aware Margins | 60.44 | **89.51** | 51.07 | 47.77 |
> | -Difficulty-aware Margins | 61.22 | 89.12 | 54.16 | **48.54** |
> | Baseline | 41.42 | 46.83 | 12.96 | 46.40 |
>
> **Q4: Parameter Sensitivity.**
>
> **Table 2 reports MMCD’s performance over a wide range of hyperparameter combinations, revealing that accuracy varies by no more than 1.07%.** This negligible fluctuation demonstrates MMCD’s strong resilience to hyperparameter selection, thereby minimizing the need for extensive tuning. Moreover, such stability indicates that MMCD can be deployed across new VQA benchmarks and domains with little to no reconfiguration. **In summary, MMCD’s insensitivity to parameter settings both streamlines practical deployment and validates its robust generalization in mitigating multimodal bias.**
>
> Table 2: Comparison of Accuracy on the VQA-CP v2 dataset with different parameter configurations.
> |λ_1| 0.01 | 0.05 | 0.1 | 0.15 | 0.2 |
> | -------- | -------- | -------- | -------- | -------- | -------- |
> |Accuracy(%)| 60.27 | 60.82 | **61.34** | 60.90 | 60.85 |
>
> |λ_2| 0.01 | 0.025 | 0.05 | 0.075 | 0.1 |
> | -------- | -------- | -------- | -------- | -------- | -------- |
> |Accuracy(%)| 61.23 | 60.87 | **61.34** | 60.77 | 61.14 |
>
> |α| 0.4 | 0.5 | 0.6 | 0.7 | 0.8 |
> | -------- | -------- | -------- | -------- | -------- | -------- |
> |Accuracy(%)| 61.11 | 60.95 | **61.34** | 60.73 | 60.82 |
>
> Finally, **we hope these clarifications enhance our work’s visibility and positively impact its evaluation scores.**

---

> > ### Comment · Reviewer_UEGk · 2025-08-06
> >
> > Thank the authors for their responses. I am overall satisfied with the rebuttal and have adjusted my score accordingly.

---

> > > ### Author Response · Authors · 2025-08-07
> > >
> > > We sincerely appreciate your recognition, the valuable adjustments made, and the sustained effort you have devoted throughout this process. Your support plays a vital role in refining and advancing our work.

---

### Official Review · Reviewer_GABY · 2025-07-16

**Clarity:** 3
**Significance:** 3
**Originality:** 3
**Rating:** 5
**Confidence:** 4

**Summary:**

This paper focuses on the issue of language bias in Visual Question Answering (VQA). The authors first conduct a thorough theoretical investigation into the origin of language bias, identifying modality-specific gradient imbalances, biased feature fusion, and classifier weight deviations as key factors. To tackle these issues, they propose a Multi-Margin Collaborative Debiasing (MMCD) framework, which consists of Multi-grained Adaptive Margins (MAM), including frequency-aware, confidence-aware, and difficulty-aware margins. And together with a Difficulty-aware Contrastive Learning (DCL) module. Extensive experiments on OOD benchmarks demonstrate that MMCD significantly outperforms previous state-of-the-art methods, showcasing its robustness and effectiveness in mitigating language bias.

**Questions:**

1. The MMCD hyperparameters (λ₁, λ₂, λ₃, α, β, and w) appear to play critical roles in the overall performance. However, it remains unclear how sensitive the method is to these hyperparameters, and how the specific combination was determined during experimentation. Can the method maintain robust performance across different datasets without extensive hyperparameter tuning?

**Ethical Concerns:**

["NO or VERY MINOR ethics concerns only"]

**Final Justification:**

My concerns have been addressed

**Limitations:**

yes

**Quality:**

3

**Strengths And Weaknesses:**

Strength
1. The paper conducts a comprehensive theoretical and empirical study into the origin of language bias in VQA, identifying three key factors, which is useful to the task and future research.
2. The proposed framework is well motivated by prior observations and insights. It integrates multi-granularity margin design (answer frequency, model confidence, and sample difficulty) with contrastive learning in a principled way.
3. The method consistently outperforms baselines on OOD benchmarks and exhibits strong generalization across diverse backbone architectures.

Weakness
1. The adaptive margin computation (particularly the sample-specific difficulty computation) likely adds complexity, but no analysis of training time or memory cost compared to baselines. This could also limit the scalability to larger datasets due to computational overhead.
2. The narrow focus on specific types of bias and the reliance on dataset statistics may significantly hinder real-world deployment.

---

> ### Author Rebuttal · Authors · 2025-07-31
>
> Thanks for your valuable comments.
>
> **Q1: Clarification of Computational Cost.**
>
> In our work, ***all experimental settings adhere to standard VQA protocols and incur only modest computational overhead.*** During training, our MAM and DCL act solely as loss‐level regularizers to shape the feature space—no additional network branches or contrastive modules are introduced. At inference, the model follows the conventional VQA forward pass without invoking any auxiliary components, leaving latency, FLOPs, and memory usage unchanged relative to the baseline. As summarized in Table 1, training FLOPs increase marginally from 0.18 G (COB) to 0.19 G (MMCD), and parameters grow from 35.18 M to 35.9 M. **Thus, compared to existing baselines, MMCD achieves significant robustness improvements with negligible inference‐time overhead and an acceptable increase in training cost.**
>
>
> **Table 1: Comparasion of FLOPs and Parameters on VQA-CP v2.**
> | Methods | Ref. | FLOPs(G) | Parameters(M) |
> | -------- | -------- | -------- | -------- |
> | UpDn | CVPR'18 | 0.18 | 12.94 |
> | RMLVQA | CVPR'23 | 0.17 | 18.88 |
> | GGD | TPAMI'23 | 0.19 | 33.98 |
> | COB | WACV'23 | 0.16 | 35.18 |
> | PHOH | AAAI'25 | 0.31 | 18.88 |
> | **MMCD** | Ours | 0.19 | 35.90 |
>
> **Q2: The Robustness of the Method.**
>
> Table 2 reports a detailed assessment of MMCD on the VQA-CE benchmark, which highlights **challenging multimodal bias patterns** in VQA. MMCD consistently outperforms baseline bias-mitigation methods across all biased test splits, delivering markedly higher accuracy. **These results demonstrate that MMCD produces stable, unbiased predictions under varied bias conditions, underscoring its superior effectiveness and robustness compared to existing approaches targeting other bias types.**
>
> **Table 2: Accuracy comparison on VQA-CE dataset.**
> | Methods | Ref. | Overall | Counter | Easy |
> | -------- | -------- | -------- | -------- | -------- |
> | CSS | CVPR'20 | 53.55 | 34.36 | 62.08 |
> | GENB | CVPR'23 | 57.87 | 34.80 | 68.15 |
> | RMLVQA | CVPR'23 | 58.08 | 35.01 | 68.21 |
> | CVIV | TMM'24 | - | 36.12 | - |
> | PHOH | AAAI'25 | 59.10 | 36.21 | 68.31 |
> | MMCD | Ours | **67.86** | **38.69** | **80.83** |
>
> **Q3: Parameter Sensitivity.**
>
> **Table 3 presents MMCD's performance across a wide spectrum of hyperparameter settings, showing that accuracy varies by at most 1.07%.** This negligible fluctuation underscores MMCD’s robustness to hyperparameter choices, thereby minimizing the need for extensive tuning. Furthermore, such stability indicates that MMCD can be seamlessly applied to new VQA benchmarks or domains without significant reconfiguration. **Overall, this insensitivity to parameter settings both streamlines deployment and validates MMCD's strong generalization in mitigating multimodal bias.**
>
>
> **Table 3: Comparison of Accuracy on the VQA-CP v2 dataset with different parameter configurations.**
> |λ_1| 0.01 | 0.05 | 0.1 | 0.15 | 0.2 |
> | -------- | -------- | -------- | -------- | -------- | -------- |
> |Accuracy(%)| 60.27 | 60.82 | **61.34** | 60.90 | 60.85 |
>
> |λ_2| 0.01 | 0.025 | 0.05 | 0.075 | 0.1 |
> | -------- | -------- | -------- | -------- | -------- | -------- |
> |Accuracy(%)| 61.23 | 60.87 | **61.34** | 60.77 | 61.14 |
>
> |α| 0.4 | 0.5 | 0.6 | 0.7 | 0.8 |
> | -------- | -------- | -------- | -------- | -------- | -------- |
> |Accuracy(%)| 61.11 | 60.95 | **61.34** | 60.73 | 60.82 |
>
> Finally, **we sincerely hope the above clarification would enhance the recognition and rating score of our work.**

---

> > ### Comment · Reviewer_GABY · 2025-08-06
> >
> > I appreciate the responses and the additional experiments from the authors. My concerns have been addressed and I have no more questions.

---

> > > ### Author Response · Authors · 2025-08-07
> > >
> > > We are glad to hear that your concerns have been effectively resolved. We truly appreciate your recognition and the thoughtful, constructive input you have provided. Your feedback has significantly contributed to enhancing the clarity, rigor, and overall quality of our work, and we remain grateful for your engagement throughout the process.

---

### Official Review · Reviewer_GGBZ · 2025-07-22

**Clarity:** 4
**Significance:** 3
**Originality:** 3
**Rating:** 5
**Confidence:** 4

**Summary:**

This paper tackles the persistent issue of language bias in Visual Question Answering (VQA), where models tend to rely on superficial linguistic correlations rather than genuine multimodal reasoning. The authors conduct a principled analysis to uncover how such biases arise and introduce a novel mitigation framework: Multi-Margin Collaborative Debiasing (MMCD). The paper makes a substantial conceptual and empirical contribution to the VQA field by clarifying the origins of language bias and proposing a theoretically grounded approach to address it.

**Questions:**

1.How sensitive is MMCD to the choice of hyperparameters (e.g., λ₁, λ₂, σ in the margin formulations), and how does this affect its generalizability across different datasets or domains?
2.Given that the difficulty-aware module relies on handcrafted metrics like learning speed and classification margin, have the authors considered replacing or complementing this with a learnable, data-driven difficulty estimator?
3.Can the proposed MMCD framework be effectively adapted to or integrated with large-scale pre-trained multimodal models (e.g., BLIP-2, Flamingo, or LLaVA), and if so, what challenges are anticipated in such a transfer?

**Ethical Concerns:**

["NO or VERY MINOR ethics concerns only"]

**Final Justification:**

My concerns have been addressed

**Limitations:**

Yes

**Quality:**

4

**Strengths And Weaknesses:**

Strengths：
1. Theoretical Insight into Bias Formation
The paper offers a principled and detailed analysis of how language bias emerges in VQA models, linking it to modality-specific gradient imbalances, feature fusion asymmetries, and classifier weight directionality. This goes beyond empirical fixes and provides a deeper understanding of the underlying mechanisms—something that has been largely underexplored in prior work.
2. Novel and Integrated Debiasing Framework (MMCD)
Despite the presence of missing data during training or inference, SHIFT maintains competitive or superior predictive performance (as measured by c-index) compared to strong baselines. The model’s generalizability across cohorts (TCGA, CPTAC, UCI) suggests it is not overfitted and can scale across populations and institutions without manual adjustment.
3. Strong Empirical Results and Generalization Across Architectures
MMCD demonstrates state-of-the-art performance on challenging datasets such as VQA-CP v2 and v1, particularly in bias-prone categories (e.g., "yes/no", "number"). The method is also shown to be model-agnostic, achieving consistent improvements across multiple backbone architectures, including SAN, S-MRL, and LXMERT.

Weaknesses：
1．The proposed MMCD framework introduces multiple auxiliary components—such as adaptive margins, difficulty tracking, and contrastive learning—which significantly increase the training complexity and computational cost. This may limit the practical applicability of the method, especially in resource-constrained or real-time scenarios.
2. While the paper provides a thorough treatment of language bias, it does not explicitly address other forms of bias prevalent in VQA, such as visual shortcut bias, question-type bias, or multimodal co-adaptation. This narrow focus may limit the generality and robustness of the proposed method when applied to broader or more complex bias scenarios.

---

> ### Author Rebuttal · Authors · 2025-07-31
>
> Thanks for your insightful comments.
>
> **Q1: Clarification of Computational Cost.**
>
> In our work, ***all experimental configurations conform to standard VQA practice and entail a modest, operationally acceptable computational footprint.*** Specifically, the proposed MAM and DCL operate exclusively during training as loss-level regularizers that shape the representation space, without invoking any auxiliary branches or contrastive operations. In the test phase, the model follows the standard VQA forward path without invoking any auxiliary branches or contrastive operations. Consequently, inference-time latency, FLOPs, and memory usage remain effectively unchanged relative to the baseline, ensuring predictable and efficient deployment. As shown in Table 1, the training FLOPs increase from only marginally 0.18G (COB) to 0.19G (MMCD) while the parameter count is listed as 35.18M (COB) vs. 35.9M (MMCD). ***Compared with the existing baselines, MMCD delivers robustness gains with negligible inference-time overhead and an acceptable training-time cost.***
>
>
> Table 1: Comparasion of FLOPs and Parameters on VQA-CP v2.
> | Methods | Ref. | FLOPs(G) | Parameters(M) |
> | -------- | -------- | -------- | -------- |
> | UpDn | CVPR'18 | 0.18 | 12.94 |
> | RMLVQA | CVPR'23 | 0.17 | 18.88 |
> | GGD | TPAMI'23 | 0.19 | 33.98 |
> | COB | WACV'23 | 0.16 | 35.18 |
> | PHOH | AAAI'25 | 0.31 | 18.88 |
> | **MMCD** | Ours | 0.19 | 35.90 |
>
>
> **Q2: The Robustness of the Method.**
>
> In Table 2, we present a comprehensive evaluation of the proposed MMCD method on the VQA-CE dataset, which emphasizes **challenging multimodal bias patterns** in VQA. MMCD achieves consistently higher accuracy across the biased test splits, outperforming baseline bias-mitigation techniques by a substantial margin. Experimental results confirm that MMCD yields stable, unbiased predictions under diverse bias scenarios. **These findings underscore the effectiveness and robustness of MMCD in addressing multimodal bias, surpassing existing methods designed for other bias types.**
>
> Table 2: Accuracy comparison on VQA-CE dataset.
> | Methods | Ref. | Overall | Counter | Easy |
> | -------- | -------- | -------- | -------- | -------- |
> | CSS | CVPR'20 | 53.55 | 34.36 | 62.08 |
> | GENB | CVPR'23 | 57.87 | 34.80 | 68.15 |
> | RMLVQA | CVPR'23 | 58.08 | 35.01 | 68.21 |
> | CVIV | TMM'24 | - | 36.12 | - |
> | PHOH | AAAI'25 | 59.10 | 36.21 | 68.31 |
> | MMCD | Ours | **67.86** | **38.69** | **80.83** |
>
>
> **Q3: Parameter Sensitivity.**
>
> In Table 3, we systematically evaluate MMCD under a comprehensive range of hyperparameter settings. **Across all combinations, accuracy fluctuates by no more than 1.07%.** Such minimal variance highlights MMCD's robustness to hyperparameter selection, significantly reducing the need for exhaustive tuning. Moreover, this stability suggests that MMCD can be readily transferred to new VQA benchmarks or application domains without extensive reconfiguration. **Overall, the insensitivity to parameter settings not only simplifies deployment but also confirms MMCD's strong generalization capability in mitigating multimodal bias.**
>
> Table 3: Comparison of Accuracy on the VQA-CP v2 dataset with different parameter configurations.
> |λ_1| 0.01 | 0.05 | 0.1 | 0.15 | 0.2 |
> | -------- | -------- | -------- | -------- | -------- | -------- |
> |Accuracy(%)| 60.27 | 60.82 | **61.34** | 60.90 | 60.85 |
>
> |λ_2| 0.01 | 0.025 | 0.05 | 0.075 | 0.1 |
> | -------- | -------- | -------- | -------- | -------- | -------- |
> |Accuracy(%)| 61.23 | 60.87 | **61.34** | 60.77 | 61.14 |
>
> |α| 0.4 | 0.5 | 0.6 | 0.7 | 0.8 |
> | -------- | -------- | -------- | -------- | -------- | -------- |
> |Accuracy(%)| 61.11 | 60.95 | **61.34** | 60.73 | 60.82 |
>
>
> **Q4: Difficulty Estimator.**
>
> Inspired by your suggestion, we propose a learnable weighted class-prototype network as a sample-difficulty estimator. Specifically, we define difficulty as the distance between each sample and its corresponding class prototype. During training, this difficulty score adaptively weights the loss, while each sample’s loss in turn updates the prototype. As a result, easy samples are gradually drawn closer to their prototypes and harder samples are repelled further away. This strategy yields rapid, accurate difficulty estimation with very low computational and memory overhead. We will implement and evaluate this approach in future work, recognizing that fast, precise difficulty estimation remains an open and compelling challenge.
>
>
> **Q5: Integration with MLLM.**
>
> To evaluate MMCD's compatibility with state-of-the-art multimodal architectures, we incorporated it into the LXMERT framework. Table 4 summarizes the comparative results: MMCD-enhanced LXMERT achieves a 18.29% absolute increase in overall accuracy on the VQA-CP v2 benchmark. **These preliminary findings demonstrate that our approach can be seamlessly adapted to large-scale transformers without modifying their core attention or fusion modules.** We expect that integrating MMCD into next-generation architectures  (e.g., BLIP-2, Flamingo, or LLaVA) will yield equal or superior benefits, owing to their richer cross-modal representations. Due to resource and scheduling constraints, we reserve these evaluations for future work, where we will also explore optimal insertion points and fine-tuning protocols to maximize gains across diverse VQA datasets.
>
> **Table 4: Performance of our approach with different network architecture.**
>
> | Methods | All | Y/N | Num | Other | Increased |
> | -------- | -------- | -------- | -------- | -------- | -------- |
> | LXMERT | 48.66 | 47.49 | 22.24 | **56.52** | - |
> | LXMERT + MMCD | **66.95** | **91.79** | **63.28** | 54.39 | **18.29** |
>
>
> Thanks! **We hope these clarifications satisfactorily resolve your questions and meet with your positive appraisal.**

---

### Decision · Program_Chairs · 2025-09-17

**Decision:**

Accept (poster)

**Comment:**

This work studied language bias in Visual Question Answering (VQA), where models over-rely on language bias rather than multimodal reasoning. The primary motivation is a lack of principled understanding of bias formation. This work conducts theoretical and empirical analyses, identifying modality-specific gradient imbalances as the core challenge. This work further proposed Multi-Margin Collaborative Debiasing (MMCD) framework with a difficulty-aware contrastive learning mechanism to dynamically determine decision boundaries. Experiments on multiple VQA benchmarks validated the effectiveness and robustness of the proposed method.

The main strengths are (1) the theoretical contributions for language bias in VQA in aspects with modality-specific gradient imbalances, skewed feature fusion, and classifier weight deviations, (2) the novelty of the proposed Multi-Margin Collaborative Debiasing (MMCD) in a principled manner, (3) strong experimental results on VQA benchmarks. The main weaknesses were (1) the complexity and computation due to multiple auxiliary modules and computational efficiency analysis, (2) generalizability to broader bias scenarios beyond language bias in VQA. After rebuttal, the reviewers' concerns have been addressed, and this work received four 5 ratings. The AC agreed with reviewers and recommended accepting this paper.